# A Tale of Two Sisters: The Southerner *Pinna rudis* Is Getting North after the Regional Extinction of the Congeneric *P. nobilis* (Mollusca: Bivalvia)

Alice Oprandi [1,*], Stefano Aicardi [2], Annalisa Azzola [1,3], Fabio Benelli [4], Marco Bertolino [1,3], Carlo Nike Bianchi [1,5], Mariachiara Chiantore [1,3], Maria Paola Ferranti [1], Ilaria Mancini [1], Andrea Molinari [6], Carla Morri [1,5] and Monica Montefalcone [1,3]

1   DiSTAV (Department of Earth, Environment and Life Sciences), University of Genoa, Corso Europa 26, 16132 Genova, Italy; annalisa.azzola@edu.unige.it (A.A.); marco.bertolino@unige.it (M.B.); carlonike.bianchi.ge@gmail.com (C.N.B.); mariachiara.chiantore@unige.it (M.C.); mariapaola.ferranti@gmail.com (M.P.F.); ilaria.mancini@edu.unige.it (I.M.); carla.morri.ge@gmail.com (C.M.); monica.montefalcone@unige.it (M.M.)
2   Sub Tribe ASD, Piazzale Kennedy 1, 16129 Genova, Italy; stefano.aicardi94@libero.it
3   NBFC (National Biodiversity Future Center), Piazza Marina 61, 90133 Palermo, Italy
4   Fabio Benelli Filmmaking, Via Oliveto 1/8, 16148 Genova, Italy; info@fabiobenelli.com
5   Genoa Marine Centre, EMI (Department of Integrative Marine Ecology), Stazione Zoologica Anton Dohrn—National Institute of Marine Biology, Ecology and Biotechnology, Villa del Principe, Piazza del Principe 4, 16126 Genova, Italy
6   OLPA (Osservatorio Ligure Pesca e Ambiente) Scrl, Via Malta 2/8, 16121 Genova, Italy; andrea.molinari1969@gmail.com
*   Correspondence: alice.oprandi@edu.unige.it

**Abstract:** In the Mediterranean Sea, the bivalve genus *Pinna* is represented by two species: the endemic *Pinna nobilis* and the (sub)tropical Atlantic *Pinna rudis*. *P. rudis* is generally less common and mostly restricted to the warmer regions of the western Mediterranean. However, since a mass mortality event, caused by a pathogen infection, has brought *P. nobilis* to the brink of extinction, records of *P. rudis* have increased in several Mediterranean regions, where it had not been previously observed. This paper reports on the presence of several *P. rudis* individuals in the Ligurian Sea, the northernmost reach of this species in the western Mediterranean. *P. rudis* has become increasingly common between 2021 and 2023, with a total of 28 new records from seven localities along the Ligurian coast. The size of the individuals and their estimated growth rate (3.6 cm·a$^{-1}$) indicated that a recruitment event most likely took place in summer 2020, when *P. nobilis* was no longer present in the area. Our observations suggest that the recruitment success of *P. rudis* increased following the decline of *P. nobilis*. However, considering the thermophilic nature of *P. rudis*, in all likelihood, the ongoing water warming is playing a crucial role in the successful establishment of this species in the Ligurian Sea. A full understanding of the recent range expansion of *P. rudis* in the Mediterranean is far from being achieved, and whether *P. rudis* will be able to fulfil the ecological role of *P. nobilis* is difficult to predict. Large scale monitoring remains the only effective way to know about the future of Pinnids in the Mediterranean Sea.

**Keywords:** new records; range expansion; sea water warming; species replacement; Ligurian Sea; Mediterranean Sea

The large fan-shell genus *Pinna* is represented by two species in the Mediterranean Sea: the endemic *P. nobilis* Linnaeus 1758 [1], and the (sub)tropical Atlantic *P. rudis* Linnaeus 1758 [2]. *P. rudis* is recognised as the sister taxon of *P. nobilis* [3], and hybrids *nobilis* × *rudis* are also known [4]. Notwithstanding its broader geographical range, in the Mediterranean

Sea *P. rudis* is less common than *P. nobilis*, being mainly restricted to the warmest south-western regions of the basin [5]. It has never been reported in the northern and cooler Ligurian Sea [6,7].

Outside the Mediterranean, where *P. nobilis* does not occur, *P. rudis* can live indifferently on sandy, rocky and muddy substrata [8]. In the Mediterranean, the two species have different habitat preferences: *P. nobilis* inhabits sandy bottoms and seagrass meadows [9], *P. rudis* rocky bottoms [10], which may indicate the outcome of interspecific competition [11]. Occasionally, the two species have been seen to coexist, resulting in mixed populations normally dominated by *P. nobilis* [10].

*P. rudis* has received less attention than its endemic sister species: the few studies available describe its Mediterranean populations as stable despite being characterised by low densities [10]. However, as its more iconic sister [12,13], *P. rudis* has been listed in Annex II of the Bern Convention as strictly protected species and in the Barcelona Convention as threatened or endangered marine species [14].

Since 2016, *P. nobilis* populations have suffered increased mortality due to a pathogen infection, which has brought the species to the brink of extinction [15,16]. In the Ligurian Sea, where the species was widespread until at least 2015 [17], an appalling dearth of living individuals has been observed since 2018 [18]; at present, *P. nobilis* has completely disappeared from the area, as well as elsewhere [16]. Listed as critically endangered in the IUCN Red List, the species has been the subject of international conservation projects in recent years [19,20]. The search for survivors is requiring a large sampling effort, partly supported by citizen science. In this context, an unusual number of *P. rudis* individuals, apparently unaffected by pathogens [21], has been recorded in various, newly discovered locations across the Mediterranean, indicating that the species is expanding its range northward and eastward [22–25].

Occurrence data in the Ligurian Sea were gathered through citizen science or in the framework of other monitoring activities that were not specifically designed to provide information on *P. rudis* but rather to verify the presence of surviving individuals of *P. nobilis*. Species identification was performed on a morphological basis, using traits easily distinguishable in the field (Table 1). The first record of *P. rudis* in the Ligurian Sea dates back to 2021, when a single individual was spotted in the Portofino Marine Protected Area (MPA). In the following two years, further 27 new records have been obtained from seven localities along the Ligurian coast (Figure 1). *P. rudis* became increasingly common in 2022 and 2023, when 12 and 15 individuals were spotted, respectively (Table 2).

Since the data were primarily retrieved from citizen observations, which were often incomplete, information on size and depth was sometimes missing: in particular, only 17 records included size (maximum shell width). Most individuals were found on rocky bottoms or stones (Figure 2), except for two records where they occurred in meadows of *Posidonia oceanica* (Linnaeus) Delile 1813. Photographic evidence proved that *P. rudis* replaced *P. nobilis* at shallow depths on the beachrock of Borgio Verezzi (Figure 3).

**Table 1.** Morphological traits used to distinguish the two Mediterranean species of *Pinna* in the field, according to various sources [19,23,26–29].

| Trait | *P. nobilis* | *P. rudis* |
|---|---|---|
| Shell | Cuneiform, posterior margin round, arched | Triangular, posterior margin slightly squared |
| Valves | Thin, asymmetrical (angulated on one side) | Thick, rather symmetrical |
| Outer surface relief | Around 20 small radiating ribs | Well-marked 5–10 radial ribs |
| Shell ornaments | Numerous small spines in the shape of gutters | Several large, widely spaced, grooved and protuberant scales |
| Shell colour | Horny brown | Fawn brown to reddish brown |
| Shell epibionts | Numerous large species with high cover | Small encrusting species on and among scales |
| Shell opening | Linear | Wavy to flattened sinusoid |
| Mantle rim colour | Pale pink | Iridescent white |

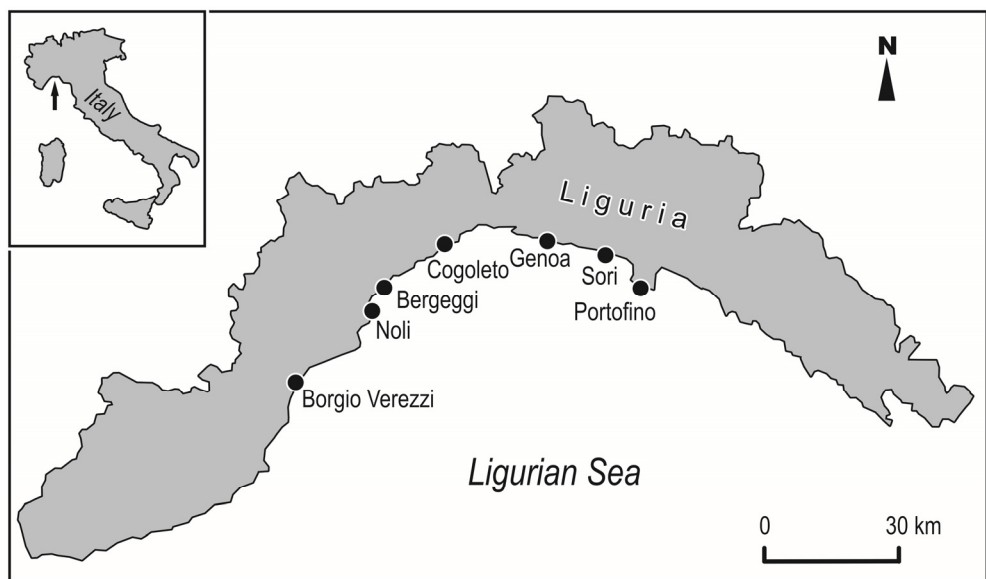

**Figure 1.** *Pinna rudis* record sites within the study area.

**Table 2.** List of *Pinna rudis* records in the Ligurian Sea in the years 2021–2023.

| Date | Site | Depth (m) | Width (cm) | Habitat | Observer |
|---|---|---|---|---|---|
| July 2021 | Portofino MPA | 16 | n.a. | Rock; semi-exposed | Claudio De Angelis |
| March 2022 | Portofino MPA | 16 | 11.5 | Rock; exposed | Alice Oprandi |
| June 2022 | Portofino MPA | 12 | n.a. | Rock; exposed | Stefano Aicardi |
| June 2022 | Portofino MPA | 7 | n.a. | Seagrass meadow; semi-exposed | Alice Oprandi |
| June 2022 | Portofino MPA | 10 | n.a. | Stones; semi-exposed | Marco Bertolino |
| July 2022 | Portofino MPA | 18 | 10 | Rock; exposed | Claudio De Angelis |
| July 2022 | Portofino MPA | 15 | n.a. | Rock; exposed | Claudio De Angelis |
| August 2022 | Bergeggi MPA | n.a. | 7 | Rock; exposed | Julian Ivaldi |
| August 2022 | Portofino MPA | 15.6 | 10 | Rock; exposed | Claudio De Angelis |
| August 2022 | Noli | n.a. | 7 | Stones; sheltered | Julian Ivaldi |
| August 2022 | Genoa | 4 | n.a. | Rock; sheltered | Carlo Nike Bianchi |
| September 2022 | Portofino MPA | 16.5 | 8 | Rock; exposed | Carlo Nike Bianchi |
| October 2022 | Genoa | 7 | 10 | Rock; sheltered. (Empty shell) | Carlo Nike Bianchi |
| February 2023 | Genoa | 7 | 9 | Rock; sheltered | Alice Oprandi |
| April 2023 | Portofino MPA | n.a. | 12 | Rock; exposed | Stefano Aicardi |
| June 2023 | Genoa | 10 | 12 | Rock; sheltered | Marco Beghi |
| August 2023 | Portofino MPA | 25 | 20 | Rock; exposed | Claudio De Angelis |
| August 2023 | Portofino MPA | 18.5 | n.a. | Rock; exposed | Claudio De Angelis |
| August 2023 | Portofino MPA | 24 | 15 | Rock; exposed | Carlo Nike Bianchi |
| August 2023 | Sori | 4 | n.a. | Rock; sheltered | Marco Bertolino |
| September 2023 | Portofino MPA | 16 | 16 | Rock; exposed | Giorgio Barsotti |
| September 2023 | Bergeggi MPA | 12 | 14 | Rock; exposed | Giacomo Gennaro |
| September 2023 | Bergeggi MPA | 8 | 11 | Seagrass meadow; semi-exposed | Giacomo Gennaro |
| September 2023 | Bergeggi MPA | 7 | 10 | Rock; exposed | Giacomo Gennaro |
| September 2023 | Cogoleto | 10 | n.a. | Rock, stones; exposed | Federica Deriu |
| September 2023 | Bergeggi MPA | 20 | n.a. | Rock, stones; semi-exposed | Stefano Pavone |
| September 2023 | Bergeggi MPA | n.a. | n.a. | Rock, stones; exposed | Stefano Pavone |
| October 2023 | Borgio Verezzi | 2 | 10 | Beachrock, stones; sheltered | Andrea Molinari |

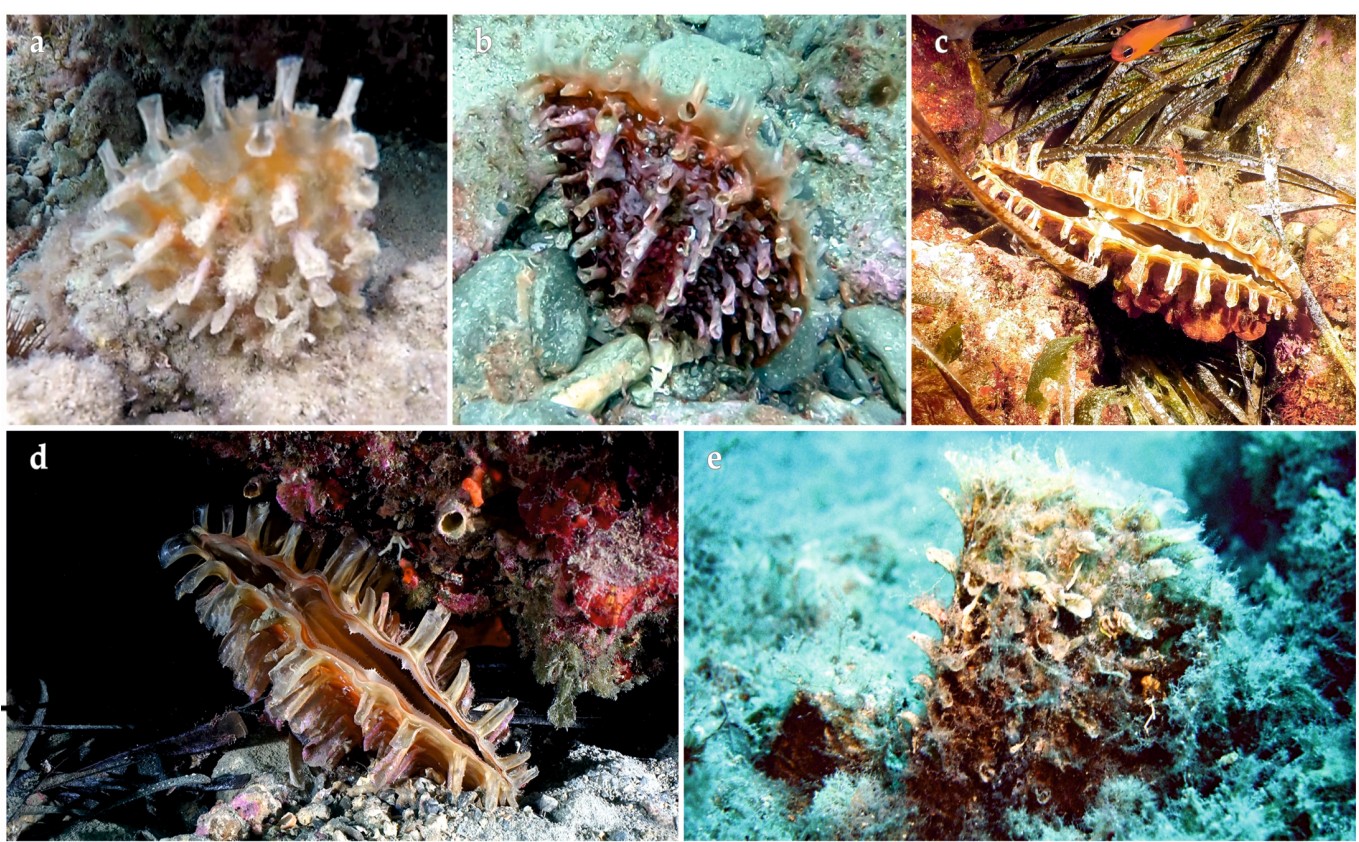

**Figure 2.** *Pinna rudis* in Bergeggi MPA (**a**) (Photo credits: J. Ivaldi), Genoa (**b**) (Photo credits: A. Oprandi), and Portofino MPA (**c**) (Photo credits: G. Galletta) (**d**) (Photo credits: G. Radicella) (**e**) (Photo credits: G. Barsotti).

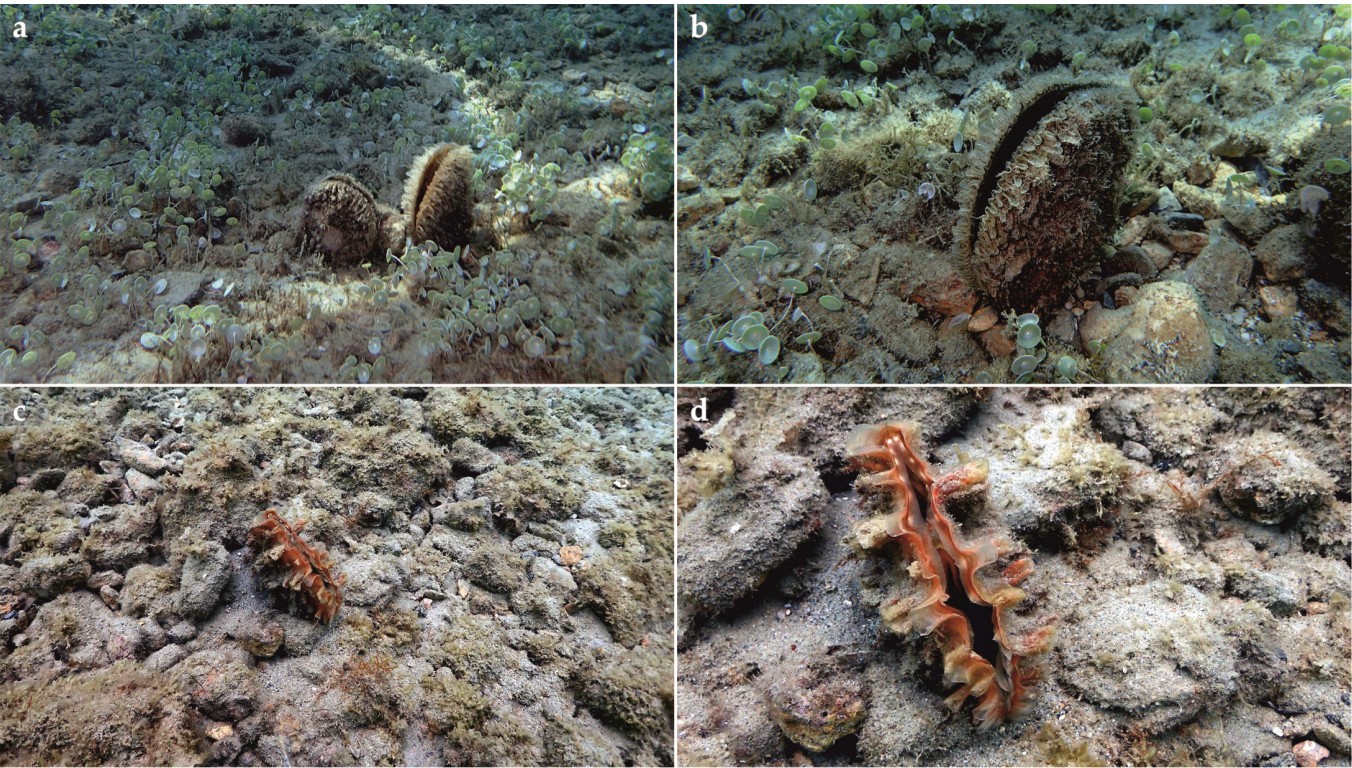

**Figure 3.** *Pinna nobilis* on the Borgio Verezzi beachrock in 2017 (**a**,**b**) replaced by *P. rudis* in 2023 (**c**,**d**) (Photo credits: A. Molinari).

The average size of *P. rudis* individuals found in the Ligurian Sea was $9.1 \pm 0.7$ cm in 2022 and $12.7 \pm 1.1$ cm in 2023 (maximum shell width), thus suggesting an average growth rate of 3.6 cm·a$^{-1}$ and a 28% increase in size (Figure 4a). Such a growth rate is slightly smaller than the 4–10 cm·a$^{-1}$ observed in SW Spain [30] and distinctly greater than the 1.1 cm·a$^{-1}$ seen in Corsica [23]. By comparison, the growth rate of *P. nobilis* is 3.5–10 cm·a$^{-1}$ [31–35]. Shell growth in *P. nobilis* can be highly variable between populations [31], and even within the same population it can change with depth [36]. Temperature, hydrodynamics and the availability of food can also have a great influence on the growth rate of *P. nobilis* [35]. In both species, juveniles grow faster than adults [30,34].

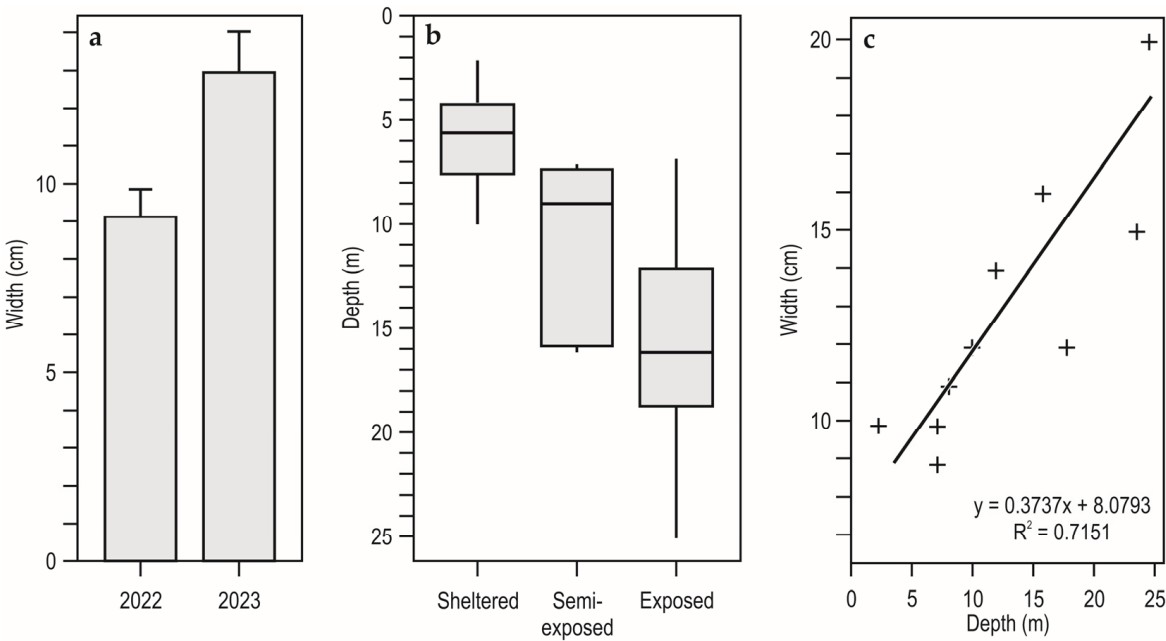

**Figure 4.** *Pinna rudis* in the Ligurian Sea: mean ($\pm$SE) shell size (maximum width) in 2022 and 2023 (**a**); preferred depth of occurrence according to exposure (**b**); relationship between shell size (maximum width) and depth in 2023 (**c**).

Only one individual was found dead in Genoa, with its empty shell still standing in a physiological position (Table 2), suggesting a low mortality rate for the species in the Ligurian Sea. The 28 record sites have been classified according to the hydrodynamic characteristics of the area in: exposed (N = 18), semi-exposed (N = 4), and sheltered (N = 6) (Table 2). A depth distribution, variable according to exposure, was observed: the more the site was exposed the deeper *P. rudis* occurred. At exposed sites, the average depth where *P. rudis* has been observed was 16.1 m, at semi-exposed sites 10.3 m and in sheltered sites 5.7 m (Figure 4b). In 2023, the individuals for which size data were available (N = 10) seemed to show a depth-related size segregation, with larger individuals at greater depths (Figure 4c); a similar trend had already been observed for *P. nobilis* in southern Spain [36].

Citizen science has been instrumental in surveying a wider geographic area and integrating scientific data on the distribution of *P. rudis*. Although untrained observers may provide inaccurate data, citizen observation proved a valuable tool. Species misidentification has been avoided as many of the individuals were photographed, allowing for proper identification through morphological traits: the occurrence of large and protuberant scales in *P. rudis*, in particular, proved the most reliable feature (Table 1). Hybrids are known to exhibit morphological characteristics intermediate between the two Mediterranean *Pinna* species; these, however, have never been observed in the Ligurian Sea. In case of doubt, molecular methods would be required for their correct identification [37]. There is a third species of Pinnidae in the Mediterranean Sea, *Atrina fragilis* (Pennant, 1777) [38], which

may also be confounded with *P. nobilis* [19]. However, *A. fragilis* is rare [38] and typically lives deeper, chiefly in the circalittoral zone [39].

Our observations suggest that the establishment of *P. rudis* in the Ligurian Sea has followed the decline of *P. nobilis*. The size of *P. rudis* individuals and their estimated growth rate are comparable to those reported in the literature [23,30], and indicate that most likely a major recruitment event took place in summer 2020, when *P. nobilis* was no longer present in the area [18]. Although *P. rudis* has a preference for cryptic habitats, which makes it difficult to detect, the obvious increase in the number of records in the Ligurian Sea suggests that it is unlikely that the species was previously present but undocumented.

As a superior competitor, it is reasonable to assume that the presence of *P. nobilis* was limiting *P. rudis* to some extent. The reasons may be competition for food, which is one of the main factors limiting growth in bivalves [30], and capture of larvae by the filtration activity of *P. nobilis* itself [10].

Undoubtedly, water warming played a crucial role in the successful establishment of *P. rudis* in the Ligurian Sea, considering the thermophilic nature of this species [5]. Since the mid-1980s, the Ligurian Sea has undergone a warming phase resulting in an increase of 1.7 °C in the yearly average temperature [40]. Several warm-water species, which were previously absent or only occasionally found in this relatively cool sea, have now successfully colonised it [41].

A full understanding of the recent range expansion of *P. rudis* in the Mediterranean is far from being achieved: it may have been favoured by the decline of its sister species, sea water warming, or both; increased incidence of a parasite and conferral of a competitive advantage to one of a pair of overlapping species are just two among the multifarious effects that temperature has on marine organisms [42]. Our observations at Borgio Verezzi (Figure 3) provide suggestive evidence that competition with *P. nobilis* may have been a limiting factor for *P. rudis* in the past.

*P. rudis* populations will probably be on the rise in the years to come. However, relict populations of *P. nobilis* have survived in sanctuary areas across the Mediterranean, mainly coastal lagoons characterised by unique and extreme physicochemical conditions [43]. Resistant individuals will eventually guarantee the recovery of *P. nobilis* in the framework of current conservation projects. Whether *P. rudis* will be able to fulfil the ecological role of *P. nobilis* is difficult to predict. Large scale and regular monitoring [44] remains the only effective way to know about the future of Pinnids in the Mediterranean Sea.

**Author Contributions:** Conceptualization, A.O. and C.N.B.; methodology, C.N.B. and C.M.; validation, A.O., M.C. and M.P.F.; formal analysis, C.N.B., C.M. and A.O.; investigation, A.O., I.M., S.A., F.B., A.A., C.N.B., M.B. and A.M.; resources, F.B., A.M. and C.M.; data curation, A.O., C.N.B., S.A., A.A., M.B. and I.M.; writing—original draft preparation, A.O. and C.N.B.; writing—review and editing, A.O., C.N.B., C.M., M.M., M.C. and M.P.F.; visualization, A.O., C.N.B. and C.M.; supervision, A.O. and M.M.; project administration, A.O.; funding acquisition, M.M. All authors have read and agreed to the published version of the manuscript.

**Funding:** This research was funded by the National Recovery and Resilience Plan (NRRP), Mission 4 Component 2 Investment 1.4—Call for tender No. 3138 of 16 December 2021, rectified by Decree n. 3175 of 18 December 2021 of the Italian Ministry of University and Research funded by the European Union—Next Generation EU; Award Number: Project code CN 00000033, Concession Decree No. 1034 of 17 June 2022 adopted by the Italian Ministry of University and Research, CUP D33C22000960007, Project title "National Biodiversity Future Center—NBFC". A.O. benefits from a research grant by the European Union LIFE PINNA project (LIFE 20 NAT/IT/00112).

**Institutional Review Board Statement:** This study did not require ethical approval.

**Data Availability Statement:** All data are presented in the present publication (see Table 1).

**Acknowledgments:** Part of the work has been carried out under the aegis of the European Union LIFE PINNA project (LIFE 20 NAT/IT/00112). The Dinghy Snipe Club (Genoa) provided logistic and field assistance for some of the surveys. Thanks are also due to Giorgio Barsotti, Claudio De Angelis and Giuseppe Galletta (GDA, Genoa), Marco Beghi, Federica Deriu, Giacomo Gennaro, Julian Ivaldi

and Stefano Pavone (Genoa), and Giovanni Radicella (Santa Margherita Ligure) for providing images and/or additional information on *Pinna rudis* in the Ligurian Sea.

**Conflicts of Interest:** The authors declare no conflicts of interest.

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
