# Peer review of "A Tale of Two Sisters: The Southerner Pinna rudis Is Getting North after the Regional Extinction of the Congeneric P. nobilis (Mollusca: Bivalvia)"

_diversity, doi:10.3390/d16020120_

Round 1
Reviewer 1 Report
Comments and Suggestions for Authors
The paper "A tale of two sisters: the southerner Pinna rudis is getting north after the regional extinction of the congeneric P. nobilis (Mollusca: Bivalvia)" by Oprandi et al. is well written and provides interesting and important information about the species Pinna rudis. Considering the great threat of the species Pinna nobilis and the disappearance of this species from a large area in the Mediterranean, finding new locations with living individuals of the species P. rudis is of great importance, especially since the individuals are alive and healthy. Any new information is welcome for a species that was not thought to be relatively rare in the Mediterranean. New data on the discovery of the species P. rudis show that perhaps the opposite is true and that the species is more common than previously thought. I have no objections to the paper, nor to the way it was written, and I fully support its publication.
Author Response
Reviewer #1 - The paper "A tale of two sisters: the southerner Pinna rudis is getting north after the regional extinction of the congeneric P. nobilis (Mollusca: Bivalvia)" by Oprandi et al. is well written and provides interesting and important information about the species Pinna rudis. Considering the great threat of the species Pinna nobilis and the disappearance of this species from a large area in the Mediterranean, finding new locations with living individuals of the species P. rudis is of great importance, especially since the individuals are alive and healthy. Any new information is welcome for a species that was not thought to be relatively rare in the Mediterranean. New data on the discovery of the species P. rudis show that perhaps the opposite is true and that the species is more common than previously thought. I have no objections to the paper, nor to the way it was written, and I fully support its publication.
Reply – We thank Reviewer #1 for their words of appreciation.
Reviewer 2 Report
Comments and Suggestions for Authors
The manuscript is adequate and good research.
Good figures and the use of adequate material and methods, which gave good result.
They must consider:
-make a table summarizing the differences between the two species
-compare, if possible, the individual growth of P. rudis estimated in this work, with theoretical knowledge of the individual growth of P. nobilis
-how to contrast the hypotheses raised in the work
Author Response
Reviewer #2 - The manuscript is adequate and good research.
Reply – We thank Reviewer #2 for their words of appreciation.
Reviewer #2 - Good figures and the use of adequate material and methods, which gave good result.
Reply – Many thanks again to Reviewer #2 for this further appreciation.
Reviewer #2 - make a table summarizing the differences between the two species
Reply – Done: see page 2, new Table 1
Reviewer #2 - compare, if possible, the individual growth of P. rudis estimated in this work, with theoretical knowledge of the individual growth of P. nobilis
Reply – Done: see page 3 lines 93-97.
Reviewer #2 - how to contrast the hypotheses raised in the work
Reply – It is difficult to imagine how to plan an experimental design to contrast the two hypothesis. Based on our field observation, however, we do believe that both sea water warming and declined competition favoured P. rudis establishment. We added a short comment in the discussion about that (see page 5 lines 148-150).
Reviewer 3 Report
Comments and Suggestions for Authors
The "Interesting Images" submission in Diversity
has to be improved:
1. According to the Life Pinna project are given the following suggestions:
Do not confuse Pinna nobilis with its sister species Pinna rudis and Atrina fragilis. https://www.lifepinna.eu/en/citizen-science-en/
So in Introduction you have to mention that there are 3 species (add the relevant references) in Ligurian of which the 2 are reported in the present study
2. As there are hybrids between P.rudis X P.nobilis is difficult to distinguish the species. So despite that there are several pictures that seems to be P. rudis, to be sure you have to mention in the discussion that further genetic identification is recommended to certify the initial phenotypic description .
Vázquez‑Luis et al.,(2021).Natural hybridization between pen shell species: Pinna rudis and the critically endangered Pinna nobilis may explain parasite resistance in P. nobilis.
Molecular Biology Reports (2021) 48:997–1004 https://doi.org/10.1007/s11033-020-06063-5
Author Response
Reviewer #3 - The "Interesting Images" submission in Diversity has to be improved:
Reply – The opinion of Reviewer # 3 strongly contradicts the opinions of the other two Reviewers, who expressed words of appreciation. However, we took fully accounts of the critics of Reviewer #3 in the revised version of the ms.
Reviewer #3 - According to the Life Pinna project are given the following suggestions: Do not confuse Pinna nobilis with its sister species Pinna rudis and Atrina fragilis. https://www.lifepinna.eu/en/citizen-science-en/ So in Introduction you have to mention that there are 3 species (add the relevant references) in Ligurian of which the 2 are reported in the present study
Reply – We are fully aware of the rationale of the Life Pinna project, as most of the authors of our paper participate in the project. As clearly stated in the title, our study concentrated on the genus Pinna, not Atrina fragilis: however, we added a short mention of Atrina fragilis as requested (see page 4 lines 118-120).
Reviewer #3 - As there are hybrids between P.rudis X P.nobilis is difficult to distinguish the species. So despite that there are several pictures that seems to be P. rudis, to be sure you have to mention in the discussion that further genetic identification is recommended to certify the initial phenotypic description.
Reply – Reviewer #3 is right. We added a sentence to highlight the problem (page 3 to 4, 115-118).
Reviewer #3 - Vázquez‑Luis et al.,(2021).Natural hybridization between pen shell species: Pinna rudis and the critically endangered Pinna nobilis may explain parasite resistance in P. nobilis. Molecular Biology Reports (2021) 48:997–1004 https://doi.org/10.1007/s11033-020-06063-5
Reply – This paper has been cited (see page 4, line 118) and the reference has been added (see page 7, lines 277-279) in the revised version of the ms.

Round 2
Reviewer 3 Report
Comments and Suggestions for Authors
SIGNIFICANT IMPROVEMENTS ARE INTRODUCED.
THE PAPER DRAFT AT THE PRESENT FORM IS WELCOME FOR PUBLICATION.